# EFFICIENT VISUAL GROUNDING VIA ALIGNMENT PRIORS AND SCALE ADAPTABILITY

## ABSTRACT

Visual Grounding links textual descriptions to the corresponding image regions, and its complexity increases with target semantic complexity. Existing methods encounter performance bottlenecks due to semantic alignment bias and scale-induced perception mismatch. In this paper, we propose ASVG, an efficient framework that exploits alignment priors from the cross-modal encoder to build target-aware queries and enhances scale adaptability through progressive cross-scale reasoning. First, we design an alignment prior-guided query generator, which embeds text-conditioned visual heatmaps into object queries to enhance their semantic discriminability. Second, we develop a progressive cross-scale decoder that builds a multi-resolution pyramid solely from single-scale features, enabling progressive cross-scale reasoning while avoiding redundant feature-pyramid fusion. In addition, we introduce a lightweight token branch and Soft Cross-head Distillation (SCD), which enforces feature consistency and adaptively reweights losses, reducing inference cost while maintaining high performance. Our method achieves significant performance gains across six VG and GREC datasets, particularly under complex or ambiguous target semantics.

## 1 INTRODUCTION

Visual Grounding (VG) aims to associate textual descriptions with visual regions by finding the referred object [Deng et al. (2021)]. Unlike object detection that relies on predefined categories, VG supports free-form textual queries to enable more flexible object localization. To overcome the single-target limitation of traditional VG, *Generalized Referring Expression Comprehension* (GREC) [He et al. (2023)] emerges as a more general paradigm. GREC extends the grounding formulation to handle textual queries that refer to single, multiple, or zero targets in an image, thereby better aligning with the complexity and ambiguity of real-world scenes.

Effective visual grounding hinges on an accurate understanding of the visual content referred to by the text, making precise cross-modal semantic alignment indispensable. Early methods focused on extending off-the-shelf object detectors. Two-stage methods first generate region proposals and then match them to the text, selecting the proposal with the highest similarity as the prediction [Zhao et al. (2024)]. In contrast, one-stage methods directly regress bounding boxes from dense anchors using fused multimodal features [Zhou et al. (2021)]. Recent works adopt stacked transformer models to explicitly capture

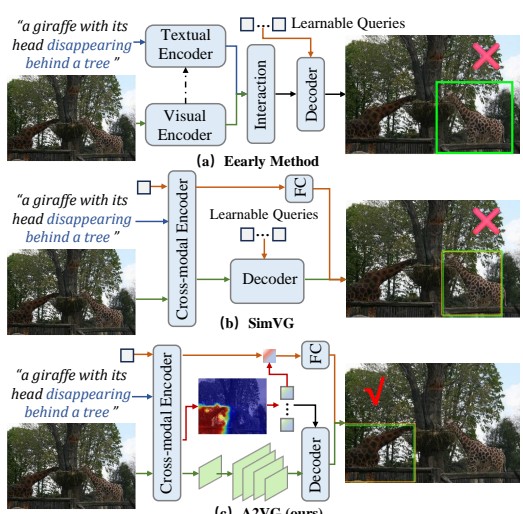

Figure 1: Qualitative comparison with prior methods. Existing methods rely on random initialization or text-guided queries and struggle in complex multimodal contexts. Our method uses alignment priors to generate discriminative queries and employs cross-scale reasoning to improve scale adaptation, enabling efficient grounding with a lightweight MLP.

cross-modal interactions [Deng et al. (2021)]. Although these methods differ in architecture, they all encode images and text independently before modality interaction, which leads to two key issues. First, such early decoupling discards latent cross-modal correlations [Deng et al. (2023)]. Second, the intrinsic distribution gap between visual and textual features must be bridged by complex interaction mechanisms [Yang et al. (2022a)]. Some studies attempt to guide image encoding with textual prompts [Su et al. (2023)], yet these prompts are still derived from textual features encoded in isolation. Moreover, these modality interaction modules are often trained from scratch on limited downstream data, which causes the learned correlations to overfit to task-specific data and limits their generalization to complex scenarios. The substantial gains observed when models are trained on mixed datasets provide indirect evidence of these issues [Shi et al. (2023)].

The rapid progress of Vision-Language Pre-trained models (VLPs) offers a promising solution to the above issues. Leveraging large-scale and heterogeneous data, VLPs establish naturally aligned cross-modal feature spaces. HiVG [Xiao et al. (2024)] and CLIP-VG [Xiao et al. (2023)] use the strong cross-modal representations of CLIP [Radford et al. (2021)] to enhance the modeling of cross-modal relations. CPT [Yao et al. (2024)] and ReCLIP [Subramanian et al. (2022)] further exploit the robust generalization ability of VLPs, extending VG into zero-shot and few-shot regimes and achieving competitive performance with minimal task-specific data. Departing from simplistic VLP embedding within existing pipelines, SimVG [Dai et al. (2024)] decouples multimodal fusion from downstream tasks and attains efficient inference by employing a two-stage training strategy with dynamic weight-balanced distillation. Although VLPs endow VG with transferable cross-modal representations and global semantic priors, two core bottlenecks remain. (1) *Semantic alignment bias*. Mainstream methods rely on learnable queries and transformer architectures to perform implicit alignment in the textual embedding space to extract semantic cues, overlooking the active role of visual evidence in semantic discrimination and disambiguation. This bias is particularly evident in complex multimodal contexts or ambiguous target semantics, resulting in unstable grounding. Recent work seeks to mitigate it by strengthening cross-modal information feedback, but typically at the cost of higher architectural and inference complexity [Wang et al. (2024b)]. (2) *Scale-induced perception mismatch*. Existing methods rely on single-scale prediction, which struggles to reconcile the details of small objects with the global structure of large ones. Multi-stage fusion can alleviate this mismatch but inevitably increases computational cost. Consequently, scale-induced perception blind spots, compounded by fusion overhead, become a bottleneck for inference performance.

In this paper, we propose ASVG, an efficient visual grounding framework built on the BEiT [Wang et al. (2023)] cross-modal encoder. To address semantic alignment bias and scale-induced perception mismatch, ASVG couples an Alignment Prior-guided Query Generator (AP-QG) with a Progressive Cross-scale Decoder (PCD). AP-QG provides target-aware guidance by injecting text-conditioned visual heatmaps into object queries, exploiting encoder-learned alignment priors to enhance their semantic discriminative power. PCD constructs a multi-resolution pyramid from single-scale features and performs progressive cross-scale reasoning, enhancing scale-aware perception while reducing redundant feature-pyramid fusion. To improve inference efficiency, we augment PCD with a parallel token branch consisting of a single linear layer and introduce Soft Cross-dead Distillation (SCD). This distillation applies feature-consistency constraints and adaptive loss reweighting, preserving PCD's high performance while substantially reducing inference overhead. Accordingly, ASVG significantly improves convergence efficiency and grounding performance by modeling text-visual semantic alignment and applying progressive cross-scale inference.

In conclusion, our main contributions are listed as follows:

- To address semantic alignment bias and scale-induced mismatch, we propose ASVG, an efficient framework that uses alignment priors to mitigate the alignment bias and employs progressive cross-scale reasoning to alleviate the mismatch.

- We introduce an Alignment Prior-guided Query Generator (AP-QG) that embeds text-conditioned visual heatmaps into object queries, exploiting encoder-learned cross-modal alignment priors to enhance their semantic discriminability.

- We propose a Progressive Cross-scale Decoder (PCD) that constructs a multi-resolution pyramid solely from single-scale features, enabling progressive reasoning across scales without resorting to complex multi-scale fusion. In addition, we introduce a token branch, coupled with Soft Cross-dead Distillation (SCD). By enforcing feature-consistency con-

Figure 2: The overall architecture of ASVG. First, AP-QG generates object queries with stronger target semantics. Next, the Progressive Cross-scale Decoder (PCD) builds a multi-resolution pyramid and performs progressive cross-scale reasoning. Then, a lightweight token branch is trained with Soft Cross-dead Distillation (SCD). This branch can be used independently at inference to increase speed.

straints and adaptive loss reweighting, SCD preserves the high performance of PCD while maintaining a lightweight advantage for the student branch.

- Extensive evaluations across six public VG and GREC datasets demonstrate that ASVG delivers superior convergence efficiency and grounding accuracy, especially in scenarios with complex or ambiguous target semantics.

## 2 RELATED WORKS

### 2.1 VISUAL GROUNDING

Visual Grounding (VG) aims to generate bounding boxes for image regions referred to by text. Existing methods fall into three main groups: two-stage methods [Yu et al. (2018); Liu et al. (2019b;a)], one-stage methods [Luo et al. (2020); Yang et al. (2019)], and transformer-based methods [Yang et al. (2022a)]. Two-stage methods separate region proposal and text matching, which enables initial cross-modal alignment. One-stage methods accelerate inference through end-to-end design, yet they still struggle with complex multimodal scenarios. The rise of transformers has introduced unified attention mechanisms that capture visual-language interactions more effectively. TransVG [Deng et al. (2021)] exemplifies this trend by capturing fine-grained semantic correspondences through token-level connections. Recent advances in Vision-Language Pre-trained models (VLPs) have further transformed VG. Large-scale aligned pretraining with models such as CLIP [Radford et al. (2021)] and BEiT-3 [Wang et al. (2023)] supplies strong semantic priors for downstream tasks. Three lines of research have emerged. Architecture-adaptation methods such as SimVG [Dai et al. (2024)] and Dynamic MDETR [Shi et al. (2023)] enhance the transfer of pretrained features through improved decoders or distillation strategies. Parameter-efficient methods such as CPT [Yao et al. (2024)] adopt prompt learning to achieve few-shot adaptation. Feature-refinement methods such as HiVG [Xiao et al. (2024)] introduce hierarchical modulation to boost grounding accuracy. These directions collectively push VG from full supervision toward few-shot settings and from coarse- to fine-granularity. In contrast, this paper focuses on exploiting the intrinsic alignment of pretrained encoders to achieve referring semantic comprehension and efficient scale adaptation.

### 2.2 KNOWLEDGE DISTILLATION

Knowledge Distillation enhances lightweight student models by guiding them to mimic a larger teacher without modifying the student architecture. Chen et al. first introduced a distillation framework for detection [Chen et al. (2017)], transferring knowledge through joint feature and prediction distillation. Later work selected informative regions to refine feature distillation [Jia et al. (2024); Dai et al. (2021)] or redesigned loss weighting strategies [Li et al. (2022); Zhixing et al. (2021)]. LD [Zheng et al. (2022)] distilled the local distribution of bounding boxes to pass spatial knowledge, whereas CrossKD [Wang et al. (2024a)] proposed cross-head distillation by routing interme-

diate student features through the teacher head to mitigate target conflicts. To improve grounding efficiency, we introduce SCD, which enforces feature consistency and adaptively reweights losses, thereby preserving the teacher decoder's accuracy while substantially reducing inference cost.

## 3 METHODOLOGY

As shown in Figure 2, our framework first jointly encodes the input image and text. Then, an Alignment Prior-guided Query Generator (AP-QG) explicitly exploits alignment priors from the encoder to generate object queries with strong target semantics. These queries are fed into the Progressive Cross-scale Decoder (PCD). In addition, a lightweight token branch is introduced, combined with Soft Cross-head Distillation (SCD), to enable more efficient inference.

### 3.1 CROSS-MODAL ENCODING

We adopt BEiT-3 [Wang et al. (2023)] as the encoder, exploiting its powerful cross-modal understanding ability to enhance image-text alignment. Specifically, the image and text inputs are transformed into token sequences via separate visual and textual embedding layers, with a learnable token serving as an explicit target representation. These tokens are then concatenated into a unified input sequence, accompanied by attention masks to guide cross-modal transformer modeling. During encoding, modality-specific parameters are independently maintained within the Feed-Forward Network (FFN), while the overall architecture adheres to the standard ViT, thus enabling effective fusion without compromising intrinsic properties. The encoder outputs are the visual token $F'_v \in \mathbb{R}^{N_v \times D_e}$, the textual token $F'_e \in \mathbb{R}^{N_e \times D_e}$, and the object token $F'_o \in \mathbb{R}^{1 \times D_e}$, where $N_v = (H/32) \times (W/32)$ (with $H$ and $W$ denoting the image height and width), and $N_e$ is the number of textual tokens.

### 3.2 ALIGNMENT PRIOR-GUIDED QUERY GENERATOR

By fusing text-conditioned visual responses with global textual semantics, AP-QG explicitly mitigates cross-modal semantic alignment bias, yielding target-aware and discriminative object queries. Specifically, two separate linear layers first project the visual token $F'_v$ and textual token $F'_e$ into a shared dimension $D_c$, yielding $F_v \in \mathbb{R}^{N_v \times D_c}$ and $F_e \in \mathbb{R}^{N_e \times D_c}$. Both token sets are then L2-normalized along the channel dimension to remove scale discrepancies across modalities, after which a fine-grained alignment matrix $M_{\text{align}} \in \mathbb{R}^{N_v \times N_e}$ is computed as follows:

$$\overline{F}_v = F_v / \|F_v\|_2 , \quad \overline{F}_e = F_e / \|F_e\|_2 , \quad M_{\text{align}} = \overline{F}_v \overline{F}_e^T, \tag{1}$$

where each entry of $M_{\text{align}}$ is the cosine similarity between corresponding pixels and words. To highlight informative words, $F'_e$ is passed through a two-layer MLP with GELU activation, and the resulting scores are normalized via softmax to produce the word attention weight $\mathcal{W}_{\text{word}} \in \mathbb{R}^{N_e \times 1}$. The weight is applied to $M_{\text{align}}$ along the $N_e$ axis, yielding a response vector $M_{\text{heatmap}} \in \mathbb{R}^{N_v \times 1}$.

$$\mathcal{W}_{\text{word}} = \mathbf{\Phi}_{\text{Softmax}}(\mathbf{\Phi}_{\text{MLP}}(F'_e)), \quad M_{\text{heatmap}} = M_{\text{align}} \mathcal{W}_{\text{word}}. \tag{2}$$

Here, $M_{\text{heatmap}}$ accentuates the visual regions most relevant to the textual description, serving as a spatial heatmap of the referred object. It is then linearly projected into a text-conditioned visual prior and fused with sentence-level semantics to produce the initial query $\mathcal{Q}_{\text{init}} \in \mathbb{R}^{D_c \times 1}$:

$$\mathcal{Q}_{\text{init}} = \mathbf{\Phi}_{\text{Linear}}(M_{\text{heatmap}}) + \mathbf{\Phi}_{\text{Max}}(F_e), \tag{3}$$

where $\mathbf{\Phi}_{\text{Max}}$ applies max pooling along the word dimension to produce a global textual vector. The resulting $\mathcal{Q}_{\text{init}}$ is replicated $N_o$ times and each copy is augmented by the learnable positional bias to form the object queries $\mathcal{Q}_{\text{object}}$ that are fed into the decoder. Semantically explicit object queries overcome the limitations of target-agnostic inference. As each query carries a distinct positional bias, the mechanism naturally generalizes to single-target, no-target, and multi-target scenarios.

### 3.3 PROGRESSIVE CROSS-SCALE DECODER

Multi-scale reasoning enhances scale-aware perception across object sizes by fusing high-resolution details with low-resolution semantic context. Existing methods use multi-stage outputs of the encoder together with an FPN [Lin et al. (2017)] for cross-scale fusion. In contrast, we introduce PCD,

which performs progressive cross-scale reasoning using only the single-scale output of the encoder. PCD consists of two stages: a lightweight pyramid construction and progressive decoding.

***Pyramid Construction***: The visual token $F_v'$ is first reshaped into a spatial feature map $\tilde{F}_v'$ of size $\frac{H}{32} \times \frac{W}{32} \times D_v$, then projected to obtain $X_{1/32} = \mathbf{\Phi}_{\text{Conv}1\times1}(\tilde{F}_v')$. Next, transposed convolutions $\mathbf{\Phi}_{\text{ConvT}}$ with kernel size and stride 2 are applied recursively to increase the feature resolution:

$$X_{1/16} = \mathbf{\Phi}_{\text{ConvT}}(X_{1/32}), \ \ X_{1/8} = \mathbf{\Phi}_{\text{ConvT}}(X_{1/16}). \tag{4}$$

This learnable upsampling acts as a data-driven interpolation scheme that approximates Shannon reconstruction in the frequency domain, allowing the network to learn low-pass filtering behaviour while adaptively restoring high-frequency details [Lim et al. (2017)]. In parallel, a $2\times2$ max-pooling operation generates $X_{1/64} = \mathbf{\Phi}_{\text{Max}}(X_{1/32})$ to retain stable global responses for large objects. Each scale-specific feature is processed with LayerNorm, followed by a $3\times3$ convolution.

$$\mathcal{Y}_s = \mathbf{\Phi}_{\text{Conv}3\times3}(\Phi_{\text{LN}}(X_s)), \ \ s \in \{1/8, 1/16, 1/32, 1/64\}, \tag{5}$$

where the convolution $\Phi_{\text{Conv}3\times3}$ approximates a discrete Laplacian operator, effectively suppressing artifacts introduced during upsampling. Finally, we obtain a set of complementary multi-resolution features $\mathcal{S} = \{\mathcal{Y}_{1/8}, \mathcal{Y}_{1/16}, \mathcal{Y}_{1/32}, \mathcal{Y}_{1/64}\}$.

***Progressive Decoding***: PCD comprises three transformer layers, each of which progressively refines the object query through bottom-up processing of adjacent-scale visual features. The computational procedure for the $i$-th layer ($i$=1,2,3) is formally defined as follows:

$$\mathcal{H}^{(i)} = \mathbf{\Phi}_{\text{Conv}3\times3}(\mathcal{Y}_{s_{i-1}} \| \mathcal{Y}_{s_i}), \ \ \mathcal{U}^{(i)} = \mathbf{\Phi}_{\text{TransL}}(\mathcal{U}^{(i-1)}, \mathcal{H}^{(i)}). \tag{6}$$

Here, $\|$ denotes channel-wise concatenation. Initially, we set $\mathcal{U}^{(0)} = \mathcal{Q}_{\text{object}}$ and $\mathcal{Y}_{s_0} = \mathcal{Y}_{1/8}$. The output from the third layer $\mathcal{U}^{(3)}$ is fed to a detection head that outputs bounding boxes $b_{\text{d}}$ and confidence scores $p_{\text{d}}$, forming the decoder prediction $\mathcal{P}_{\text{PCD}} = \{b_{\text{d}}, p_{\text{d}}\}$.

### 3.4 TOKEN BRANCH & SOFT CROSS-HEAD DISTILLATION

To enable lightweight inference while preserving the high performance of PCD, we introduce a token branch guided by Soft Cross-head Distillation (SCD). First, we fuse the object queries $\mathcal{Q}_{\text{object}}$ with the object token $F_o' \in \mathbb{R}^{1 \times D_e}$ to obtain an object token $F_o$ with enhanced target semantics.

$$F_o = \mathbf{\Phi}_{\text{Linear}}(\mathbf{\Phi}_{\text{Linear}}(F_o') + \mathcal{Q}_{\text{object}}). \tag{7}$$

***Soft Cross-head Distillation***: Prediction-level distillation conveys task-specific knowledge and is well-suited for training compact models. To avoid conflicting supervision from ground-truth annotations and teacher predictions, the object token $F_o$ is forwarded to the teacher head and the two heads are required to produce identical outputs. We further adjust the relative contributions of the VG loss components based on PCD's reasoning capability, using an adaptive loss reweighting scheme to reallocate their weights. Specifically, $F_o$ is first processed by a parameter-isolated detection head to yield the token branch output $\mathcal{P}_{\text{TB}} = \{b_{\text{tb}}, p_{\text{tb}}\}$. Then, $F_o$ is fed through the PCD's detection head to generate the cross-head distillation prediction $\mathcal{P}_{\text{SCD}} = \{b_{\text{scd}}, p_{\text{scd}}\}$.

We adopt the Hungarian matching strategy [Carion et al. (2020)], and the matching cost includes three components: Binary Cross-Entropy Loss, L1 Loss, and GIOU Loss. Let $\varpi(k)$ denote the index of the prediction matched to the $k$-th ground-truth target ($1 \leq k \leq N_q$ where $N_q$ is the total prediction count). The ground-truth is denoted as $\mathcal{G}_{\text{gt}} = \{b_{\text{gt}}, p_{\text{gt}}\}$ where $b_{\text{gt}}$ contains bounding boxes and $p_{\text{gt}}$ indicates target presence. The reasoning capability of PCD is quantified as follows:

$$\partial_{\text{pcd}} = N_g^{-1} \sum_{j=1}^{N_g} \mathbf{\Phi}_{\text{IoU}}(b_{\text{gt}}^{(j)}, b_{\text{d}}(\varpi(j))) \, \mathbf{\Phi}_{\text{Score}}(p_{\text{d}}(\varpi(j))), \tag{8}$$

where $N_g$ is the number of ground-truth targets. $\mathbf{\Phi}_{\text{Score}}$ extracts the foreground confidence from the matched prediction. Given $\partial_{\text{PCD}}$ and a threshold $\theta$, the VG loss is computed in stages. In the early stage, when the distillation signal is unstable, the VG loss is defined as:

$$\mathcal{L}_{\text{VG}} = \mathcal{L}_{\text{Det}}(\mathcal{P}_{\text{PCD}}, \mathcal{G}_{\text{gt}}) + \partial_{\text{pcd}}\mathcal{L}_{\text{Det}}(\mathcal{P}_{\text{SCD}}, \mathcal{P}_{\text{PCD}}) + \mathcal{L}_{\text{Det}}(\mathcal{P}_{\text{TB}}, \mathcal{G}_{\text{gt}}). \tag{9}$$

When $\partial_{\text{pcd}} > \theta$, it indicates that the teacher has achieved a reliable reasoning capacity. At this stage, an adaptive loss reweighting scheme is applied to allow the three loss components to compete adaptively and reallocate their contributions to the total VG loss.

$$\mathcal{L}_{\text{VG}} = \sum_{r \in \mathcal{R}} \beta_r \, \mathcal{L}_{\text{Det}}(\mathcal{P}_r, \mathcal{T}_r), \quad \mathcal{R} = \{\text{PCD, SCD, TB}\}. \tag{10}$$

$\mathcal{L}_{\text{Det}}(\mathcal{P}_r, \mathcal{T}_r)$ compares predictions $\mathcal{P}_r$ with targets $\mathcal{T}_r$, where $\mathcal{T}_{\text{PCD}} = \mathcal{G}_{\text{gt}}, \mathcal{T}_{\text{TB}} = \mathcal{G}_{\text{gt}}$ and $\mathcal{T}_{\text{SCD}} = \mathcal{P}_{\text{PCD}}$. $\mathcal{L}_{\text{Det}}$ is composed as: $\gamma_1 \mathcal{L}_{\text{CE}} + \gamma_2 \mathcal{L}_{\text{L1}} + \gamma_3 \mathcal{L}_{\text{GIOU}}$ with default weights $\gamma_1 = 1, \gamma_2 = 5$, and $\gamma_3 = 2$. The weights $\beta_r$ are obtained by a temperature-scaled softmax over the branch losses with temperature $\tau$. By design, $\partial_{\text{pcd}}$ is small early in training and grows as the teacher improves. The adaptive reweighting prevents the teacher decoder from being underweighted in later stages, ensuring stable optimization during knowledge transfer.

### 3.5 Network Training

The model is supervised by the VG loss $\mathcal{L}_{\text{VG}}$ and the heatmap alignment loss $\mathcal{L}_{\text{HA}}$. $\mathcal{L}_{\text{VG}}$ improves target grounding accuracy, while $\mathcal{L}_{\text{HA}}$ enforces spatial consistency between the predicted heatmap $M_{\text{Heatmap}}$ and the ground-truth distribution. The overall objective is a weighted sum of the two losses:

$$\mathcal{L}_{\text{Total}} = \mathcal{L}_{\text{VG}} + \lambda \mathcal{L}_{\text{HA}}, \tag{11}$$

with

$$\mathcal{L}_{\text{HA}} = \Phi_{\text{KL}}\big(\mathcal{P}_{\text{gt}} \| \mathcal{Q}_{\text{Heatmap}}\big) = \sum_{j=1}^{N_v} \mathcal{P}_{\text{gt}}^j \log(\mathcal{P}_{\text{gt}}^j / \mathcal{Q}_{\text{Heatmap}}^j). \tag{12}$$

Here, $\lambda$ is the weight for $\mathcal{L}_{\text{HA}}$. $\mathcal{P}_{\text{gt}}^j$ and $\mathcal{Q}_{\text{Heatmap}}^j$ denote the ground-truth and predicted probabilities at spatial location $j$, respectively. To construct $\mathcal{P}_{\text{gt}}$, a binary mask is first generated from the ground-truth bounding boxes at the original image resolution (object pixels set to 1, background to 0). This mask is then downsampled to the resolution of $M_{\text{Heatmap}}$ (typically $(H/32) \times (W/32)$) and normalized to sum to one. In parallel, a softmax function is applied to the predicted heatmap $M_{\text{Heatmap}}$, resulting in the predicted distribution $\mathcal{Q}_{\text{Heatmap}}$.

## 4 Experimental Results

### 4.1 Experimental Settings

***Datasets and Evaluation Metrics.*** We evaluate the effectiveness of ASVG on VG and GREC using six mainstream datasets: five VG datasets (RefCOCO/+ [Yu et al. (2016)], RefCOCOg [Nagaraja et al. (2016)], Flickr30k Entities [Plummer et al. (2015)], ReferItGame [Kazemzadeh et al. (2014)]) and one GREC dataset (gRefCOCO [He et al. (2023)]). RefCOCO+ forbids explicit spatial terms, increasing reliance on appearance cues and semantic complexity. RefCOCOg uses non-interactive annotation, yielding longer and more complex descriptions. The evaluation follows existing works [Deng et al. (2021)]. For VG, we report accuracy Acc@0.5, where a prediction is considered correct if its IoU with the ground-truth region is at least 0.5. For GREC, we report Pr@($F_1$=1, IoU$\geq$0.5) and N-acc. Pr@($F_1$=1, IoU$\geq$0.5) is the proportion of samples whose prediction is matched one-to-one to the ground truth by the highest-IoU assignment with IoU$\geq$0.5, and whose per-sample $F_1$ equals 1 ($F_1$ = 2TP/(2TP + FP + FN)). N-acc is evaluated only on no-target samples, where an empty prediction is treated as a true positive and any non-empty prediction as a false negative, and it equals TP/(TP+FN). Details of the dataset are provided in Appendix A.2.

***Implementation Details.*** All images from the datasets are resized to 640×640, and text lengths are truncated to 20 tokens. The training batch size is set to 20. ASVG is trained for 60 epochs on the VG datasets and for 240 epochs on the GREC dataset. All experiments are conducted on four NVIDIA RTX 3090 GPUs. Additional implementation details are provided in Appendix A.3.

### 4.2 Comparison with State-of-the-Art Methods

As shown in Table 1, ASVG achieves SOTA or competitive results on most splits across five VG benchmarks, with larger gains on semantically complex splits or those with a higher proportion of non-human targets, such as RefCOCO/+ testB. On datasets dominated by short noun phrases

Table 1: Experimental results on VG datasets. Best results are shown in **bold**, and second-best are underlined. ∗ indicates inference time on NVIDIA RTX 3090, other timings on GTX 1080Ti.

| Methods | Visual Encoder | RefCOCO val | testA | testB | RefCOCO+ val | testA | testB | RefCOCOg val-g | val-u | test-u | ReferIt test | Flickr30k test | Time (ms) |
|---|---|---|---|---|---|---|---|---|---|---|---|---|---|
| *Two-stage* | | | | | | | | | | | | | |
| MAttNet [Yu et al. (2018)] | RN101 | 76.40 | 80.43 | 69.28 | 64.93 | 70.26 | 56.00 | - | 66.58 | 67.27 | 29.04 | - | 320 |
| CM-Att-Erase [Liu et al. (2019b)] | RN101 | 78.35 | 83.14 | 71.32 | 68.09 | 73.65 | 58.03 | - | 67.99 | 68.67 | - | - | - |
| PBREC-MT [Zhao et al. (2024)] | RN101 | 82.94 | 86.31 | 80.81 | 74.85 | 79.53 | 65.60 | - | 73.86 | 73.86 | - | - | - |
| NMTree [Liu et al. (2019a)] | RN101 | 76.41 | 81.21 | 70.09 | 66.46 | 72.02 | 57.52 | 64.62 | 65.87 | 66.44 | - | - | - |
| *One-stage* | | | | | | | | | | | | | |
| FAOA [Yang et al. (2019)] | DN53 | 71.15 | 74.88 | 66.32 | 56.86 | 61.89 | 49.46 | - | 59.44 | 58.90 | 60.67 | 68.71 | 39 |
| ReSC$_L$ [Yang et al. (2020)] | DN53 | 77.63 | 80.45 | 72.30 | 63.59 | 68.36 | 56.81 | 63.12 | 67.30 | 67.20 | 64.60 | 69.28 | 36 |
| MCN [Luo et al. (2020)] | DN53 | 80.08 | 82.29 | 74.98 | 67.16 | 72.86 | 57.31 | - | 66.46 | 66.01 | - | - | 56 |
| RealGIN [Zhou et al. (2021)] | DN53 | 77.25 | 78.70 | 72.10 | 62.78 | 67.17 | 54.21 | - | 62.75 | 62.33 | - | - | 35 |
| *Transformer-based* | | | | | | | | | | | | | |
| TransVG [Deng et al. (2021)] | RN101 | 81.02 | 82.72 | 78.35 | 64.82 | 70.70 | 56.94 | 67.02 | 68.67 | 67.73 | 70.73 | 79.10 | 62 |
| QRNet [Ye et al. (2022)] | Swin-S | 84.01 | 85.85 | 82.34 | 72.94 | 76.17 | 63.81 | 71.89 | 73.03 | 72.52 | 74.61 | 81.95 | - |
| VLTVG [Yang et al. (2022a)] | RN101 | 84.53 | 87.69 | 79.22 | 73.60 | 78.37 | 64.53 | 72.53 | 74.90 | 73.88 | 71.60 | 79.18 | 79* |
| Dyn.MDETR [Shi et al. (2023)] | ViT-B/16 | 85.97 | 88.82 | 80.12 | 74.83 | 81.70 | 63.44 | 72.21 | 74.14 | 74.49 | 70.37 | 81.89 | - |
| TransVG++ [Deng et al. (2023)] | ViT-B/16 | 86.28 | 88.37 | 80.97 | 75.39 | 80.45 | 66.28 | 73.86 | 76.18 | 76.30 | 74.70 | | - |
| TransCP [Tang et al. (2023)] | RN50 | 84.25 | 87.38 | 79.78 | 73.07 | 78.05 | 63.35 | 72.60 | - | - | 72.05 | 80.04 | 74* |
| CLIP-VG [Xiao et al. (2023)] | ViT-B/16 | 84.29 | 84.29 | 84.29 | 69.55 | 69.55 | 69.55 | 72.64 | 72.64 | 72.64 | 70.89 | 81.99 | - |
| SimVG$_{TB}$ [Dai et al. (2024)] | ViT-L/32 | 90.61 | 92.53 | **87.68** | 85.36 | 89.61 | 79.74 | 79.34 | 85.99 | 86.83 | 79.30 | 82.61 | 101 |
| SimVG$_{DB}$ [Dai et al. (2024)] | | 90.51 | 92.37 | 87.07 | 84.88 | 88.50 | 78.66 | 80.43 | 85.72 | 86.70 | 78.75 | 83.15 | 116 |
| HiVG [Xiao et al. (2024)] | ViT-L/14 | 88.14 | 91.09 | 83.71 | 80.10 | 86.77 | 70.53 | - | 80.78 | 80.25 | 76.23 | 82.16 | - |
| ASVG-T | ViT-B/32 | 87.88 | 90.10 | 84.41 | 80.43 | 84.98 | 72.98 | 78.69 | 80.06 | 81.48 | 76.47 | 82.35 | 24.97* |
| ASVG-D | | 87.86 | 89.99 | 84.45 | 80.43 | 84.98 | 72.98 | 78.81 | 80.48 | 81.59 | 76.69 | 82.40 | 29.87* |
| ASVG-T | ViT-L/32 | 91.07 | **93.10** | **87.70** | **85.98** | 89.61 | 80.56 | **84.95** | 86.42 | 86.71 | 79.72 | 85.71 | 45.38* |
| ASVG-D | | **91.13** | 93.06 | 87.65 | 85.95 | **89.70** | **80.60** | 84.92 | **86.48** | **86.90** | **79.82** | **87.14** | 52.41* |

Table 2: GREC results on gRefCOCO with a 0.7 threshold for all methods.

| Methods | val Pr@($F_1$=1, IoU≥0.5) | N-acc. | testA Pr@($F_1$=1, IoU≥0.5) | N-acc. | testB Pr@($F_1$=1, IoU≥0.5) | N-acc. |
|---|---|---|---|---|---|---|
| MCN [Luo et al. (2020)] | 28.0 | 30.6 | 32.3 | 32.0 | 26.8 | 30.3 |
| VLT [Ding et al. (2021)] | 36.6 | 35.2 | 40.2 | 34.1 | 30.2 | 32.5 |
| MDETR [Kamath et al. (2021)] | 42.7 | 36.3 | 50.0 | 34.5 | 36.5 | 31.0 |
| UNINEXT [Yan et al. (2023)] | 58.2 | 50.6 | 46.4 | 49.3 | 42.9 | 48.2 |
| Ferret [You et al. (2024)] | 54.8 | 48.9 | 49.5 | 45.2 | 43.5 | 43.8 |
| SimVG$_{TB}$ [Dai et al. (2024)] | 61.3 | 56.1 | 61.7 | 58.0 | 53.1 | 57.5 |
| SimVG$_{DB}$ [Dai et al. (2024)] | 62.1 | 54.7 | 64.6 | 57.2 | 54.8 | 57.2 |
| ASVG-T$_{ViT-B/32}$ | 63.9 | 58.1 | 63.3 | 59.9 | 54.8 | 58.0 |
| ASVG-D$_{ViT-B/32}$ | **65.0** | **60.9** | **65.2** | **61.7** | **55.7** | **59.9** |

(ReferIt and Flickr30k), ASVG-D (ViT-B) reaches 76.69% and 82.40%, surpassing existing SOTA methods and improving further with larger encoder capacity. Relative to the RefCOCO series, these short texts are simpler yet, because of weaker context, more susceptible to semantic ambiguity. These results indicate that ASVG effectively handles the referred objects with ambiguous semantics. Unlike two-stage [Liu et al. (2019a)] and conventional one-stage methods [Zhou et al. (2021)], ASVG does not rely on region proposals or complex multi-stage fusion, which reduces error propagation and structural redundancy. Among transformer-based methods, ASVG (ViT-L) outperforms SimVG (ViT-L) by an average of 0.37% on RefCOCO, 0.52% on RefCOCO+, and 1.69% on RefCOCOg. ASVG injects the alignment priors from the pre-trained encoder into object queries and employs progressive cross-scale reasoning to enhance the model's understanding of complex multimodal scenarios. With the token branch and SCD, ASVG-T (ViT-B) maintains high accuracy while achieving lower latency, with a single-image inference time of 24.97 ms on an NVIDIA RTX 3090.

As shown in Table 2, ASVG adapts to GREC by simply increasing the number of object queries and attains comprehensive gains on gRefCOCO. ASVG-D achieves 65.0% Pr@0.5 and 60.9% N-acc. on the val set, improving over SimVG-DB by 2.9% and 6.2%. On the more challenging testB set, ASVG-D reaches 59.9% N-acc., outperforming UNINEXT [Yan et al. (2023)] by 11.7% while keeping Pr@0.5 consistently higher than SimVG-DB. These results indicate that ASVG, under a compact end-to-end design, enhances robustness and overall accuracy in generalized visual grounding through explicit alignment priors and progressive cross-scale reasoning.

## 4.3 ABLATION STUDY

***Analysis on AP-QG.*** As shown in Table 3, AP-QG delivers consistent gains over other variants for both the token branch and PCD, showing the effectiveness of explicitly embedding text-conditioned visual heatmaps into object queries. Heatmap-only and Text-only isolate the two core components of AP-QG: the former provides a spatial alignment prior, while the latter imposes a global semantic constraint. Text-only underperforms Heatmap-only on the testA and testB splits, indicating that removing visual priors weakens the performance of object queries. The full AP-QG yields larger improvements on testB, suggesting greater effectiveness on complex multimodal contexts. TQG is presented in SimVG [Dai et al. (2024)]. Experimental results indicate that incorporating alignment priors is more effective for producing discriminative queries than relying solely on text.

Table 3: Ablation of AP-QG. Each cell reports Token / Decoder accuracy (%).

| Methods | RefCOCO | | |
|---|---|---|---|
| | val | testA | testB |
| w/o AP-QG | 87.28 / 87.29 | 89.17 / 89.30 | 82.94 / 83.11 |
| Heatmap-only | 87.61 / 87.66 | 89.95 / 90.00 | 84.14 / 84.27 |
| Text-only | 87.62 / 87.82 | 89.35 / 89.70 | 83.71 / 83.81 |
| w Text-guided Query Generation (TQG) | 87.33 / 87.34 | 89.96 / 90.01 | 84.23 / 84.22 |
| w AP-QG | 87.88 / 87.86 | 90.10 / 89.99 | 84.41 / 84.45 |

Table 4: Impact of pyramid scales in PCD with scales 1/8, 1/16, 1/32, 1/64. When a scale is removed, 1/32 is used instead to keep compute constant. Each cell reports Token / Decoder accuracy (%).

| Pyramid Configuration | RefCOCOg | | | | |
|---|---|---|---|---|---|
| | val-u | small | medium | large | test-u |
| Single-scale@1/32 | 80.26 / 80.35 | 16.67 / 33.33 | 67.40 / 67.41 | 82.68 / 82.72 | 80.57 / 80.61 |
| w/o 1/8 → 1/32 | 79.40 / 79.62 | 16.67 / 16.67 | 67.24 / 67.59 | 82.70 / 82.74 | 80.55 / 80.64 |
| w/o 1/64 → 1/32 | 79.98 / 79.90 | 16.67 /16.67 | 67.27 / 68.09 | 82.83 / 82.83 | 80.67 / 80.80 |
| Full PCD | 80.06 / 80.48 | 62.50 / 62.50 | 67.80 / 68.35 | 83.89 / 83.92 | 81.48 / 81.59 |

Table 5: Impact of the token branch and SCD. Each cell reports Token / Decoder accuracy (%).

| Methods | RefCOCO | | |
|---|---|---|---|
| | val | testA | testB |
| w/o Token Branch | 0.00 / 88.26 | 0.00 / 90.07 | 0.00 / 85.46 |
| Token Branch (w/o SCD) | 87.85 / 87.92 | 89.21 / 89.42 | 84.12 / 84.67 |
| Token Branch (w SCD) | 87.88 / 87.86 | 90.10 / 89.99 | 84.41 / 84.45 |

Table 6: Ablation of Cross-Head Distillation (CHD) and Adaptive Loss Reweighting (ALR).

| CHD | ALR | RefCOCO | | |
|---|---|---|---|---|
| | | val | testA | testB |
| - | - | 87.58 / 87.70 | 89.89 / 89.98 | 83.82 / 84.13 |
| - | ✓ | 87.49 / 87.53 | 89.92 / 89.98 | 84.30 / 84.37 |
| ✓ | - | 87.67 / 87.69 | 90.02 / 89.92 | 83.71 / 83.93 |
| ✓ | ✓ | 87.88 / 87.86 | 90.10 / 89.99 | 84.41 / 84.45 |

***Ablation Study of PCD.*** To analyze scale effects, we split RefCOCOg test-u by the standard COCO area thresholds into small (area>1024), medium (1024≤area≤9216), and large (area<9216), with 6/1432/8164 samples, respectively. As shown in Table 4, Full PCD outperforms the single-scale 1/32 baseline, confirming that progressive cross-scale reasoning improves overall generalization. By object size, small objects benefit most from Full PCD. When either the 1/8 or 1/64 level is removed, performance drops to 16.67%/16.67% (Token / Decoder) or 33.33% on the Decoder. Note that the small subset contains only six samples, so the metrics change in steps of about 16.7%, yet the upward trend is clear. Overall, high-resolution levels such as 1/8 provide fine detail and low-resolution levels such as 1/64 provide the global structure. The intermediate 1/16 and 1/32 levels bridge the two, forming a complementary pyramid that benefits both the token branch and PCD.

***Impact of the Token Branch and SCD.*** As shown in Table 5, "w/o Token Branch" serves as the PCD-only baseline, where the token branch is absent and token-side metrics are 0. Adding the token branch without SCD improves token accuracy but slightly reduces PCD performance relative to the baseline. These results suggest that, while the token branch offers a lightweight path, sharing the encoder with PCD and joint training interfere with PCD learning, resulting in a slight reduction in teacher accuracy. When SCD is applied, the token branch improves further, and PCD partially recovers, though it remains slightly below the 'w/o SCD' setting. Overall, SCD enables the token branch to strengthen its discriminative ability and deliver a lower-latency inference path at a small cost in teacher accuracy. Notably, all results are obtained with one-stage online distillation, while training ASD first and then applying dual-branch distillation can further improve accuracy.

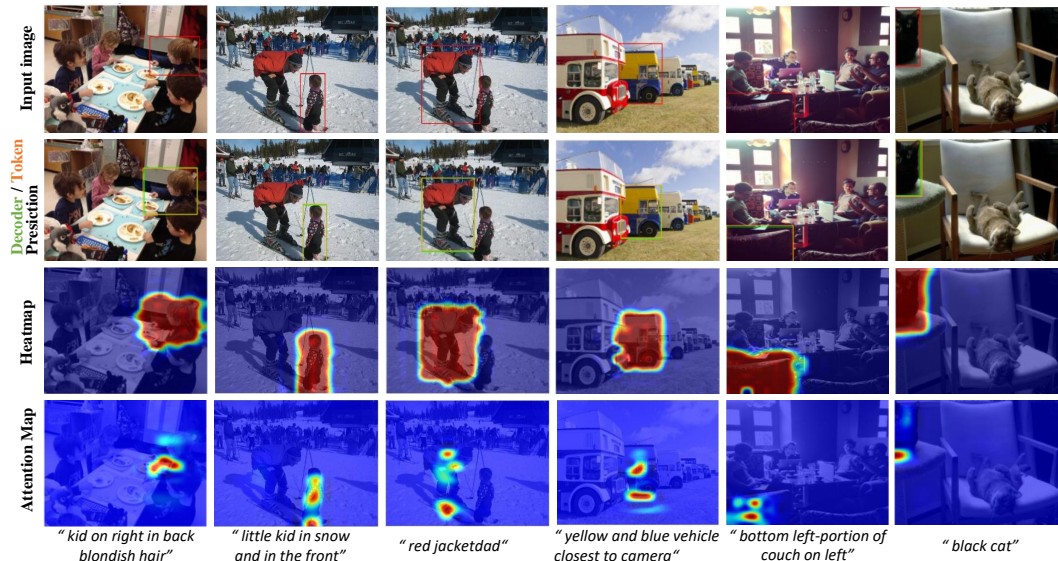

Figure 3: Visualization of model predictions, heatmaps from AP-QG, and attention maps from PCD. The red, green, and orange boxes denote ground truth, PCD predictions, and token branch predictions, respectively.

As shown in Table 6, we analyze the internal design of SCD. Introducing ALR alone yields stable gains on testB, suggesting that adaptive reweighting regularizes learning in complex scenes. However, its limited effect on val and testA indicates that ALR alone cannot fully exploit distillation. Enabling CHD alone improves the student on testA but significantly reduces performance on testB, implying that cross-head distillation benefits the student while introducing interference to the teacher. The two components are complementary: CHD offers explicit cross-head supervision, while ALR balances the contributions of different loss components through adaptive weighting.

### 4.4 QUALITATIVE RESULTS

Figure 3 presents the input image, the predictions from the PCD and token branch, the heatmap from AP-QG, and the last-layer decoder attention map. Overall, the PCD predictions (green boxes) align closely with the ground-truth annotations (red boxes), confirming stable localization. The token branch (orange boxes) is lightweight yet follows the teacher with consistent prediction trends. The text-conditioned visual responses in the heatmap highlight semantically relevant regions, as in "red jacket dad" and "yellow and blue vehicle closest to camera," indicating that AP-QG exploits alignment priors to enhance the semantic discriminability of the object query. The attention map reflects the alignment between queries and visual features during decoding. The results show that attention is primarily concentrated on specific targets, indicating that the model can further focus on key regions in complex backgrounds. More visualizations can be found in Appendix A.7.

## 5 CONCLUSION

This paper presents ASVG, an efficient VG framework that integrates alignment priors with cross-scale reasoning to enhance the semantic discriminability of queries and adapt to targets of varying scales. First, we propose an Alignment Prior-guided Query Generator (AP-QG), which explicitly exploits the cross-modal alignment prior from the encoder to produce target-aware object queries. Second, we design a Progressive Cross-scale Decoder (PCD) that builds a multi-resolution pyramid from single-scale features, enabling progressive reasoning across scales without redundant feature-pyramid fusion. To further improve efficiency, we introduce a lightweight token branch with Soft Cross-head Distillation (SCD) that enforces feature consistency and adaptively reweights losses, reducing inference cost while maintaining accuracy. Extensive experiments on six popular VG and GREC datasets demonstrate the effectiveness of ASVG.

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

## A    APPENDIX

We provide an overview of the Appendix below:

- Appendix A.1: Use of LLMs.
- Appendix A.2: Dataset Descriptions.
- Appendix A.3: Additional Implementation Details.
- Appendix A.4: Comparison with Pre-trained Models.
- Appendix A.5: Additional Ablation Studies.
    - Ablations for Object Query Generation.
    - Effect of Hyperparameters in Adaptive Loss Reweighting.
    - Analysis of the Heatmap Alignment Loss.
- Appendix A.6: Training Efficiency.
- Appendix A.7: Supplementary Qualitative Results.

### A.1    USE OF LLMS

Large Language Models (LLMs) are used to assist in improving the clarity, fluency, and grammar of the English writing in this paper. They help refine sentence structure and word choice to ensure better readability. LLMs are not involved in any aspect of the research ideation, technical development, model design, experimentation, or result interpretation. Their use is limited to language refinement and does not constitute a scientific contribution.

### A.2    DATASET DESCRIPTIONS

We evaluate the proposed ASVG on six mainstream VG and GREC datasets. Table 7 summarizes the basic statistics and split details of each dataset.

- **ReferitGame (ReferIt)** [Kazemzadeh et al. (2014)], introduced in 2014, is the first large-scale real-world dataset for VG. It is originally designed for the Phrase Grounding (PG) task and is later widely adopted in VG benchmarks. Textual descriptions are collected through an online two-player game, resulting in colloquial and task-oriented queries.

- **RefCOCO and RefCOCO+** [Yu et al. (2016)] both contain two test splits, testA and testB, where testA includes only people-related annotations and testB contains all other objects. Their textual descriptions are also collected through the interactive game mechanism used in ReferIt. Unlike RefCOCO, RefCOCO+ prohibits explicit spatial or directional terms, thus emphasizing visual appearance and enriching textual descriptions.

- **RefCOCOg** [Nagaraja et al. (2016)] contains descriptions collected via non-interactive annotation sessions on Amazon Mechanical Turk, producing longer and more syntactically complex text. Two common splits are RefCOCOg-google and RefCOCOg-umd. The google split has no public test set and contains overlaps between the training and validation sets, whereas the umd split has no overlap. To avoid potential data leakage and following prior studies, we excluded the google split in our mixed-dataset pretraining setting.

- **Flickr30k Entities (Flickr30k)** [Plummer et al. (2015)] targets the PG task. Its descriptions are noun phrases extracted from image captions rather than complete sentences, and a single caption may refer to multiple objects. Compared with RefCOCO/+/g, it contains shorter text with less context, resulting in higher ambiguity and greater annotation noise.

- **gRefCOCO** [He et al. (2023)] covers 60,287 distinct instances from 19,994 images, with a total of 278,232 textual descriptions, including 80,022 multi-target and 32,202 no-target descriptions. Some single-target descriptions are inherited from RefCOCO [Yu et al. (2016)]. gRefCOCO emphasizes robust modeling for complex referring expressions involving multiple targets or no-target cases.

Table 7: Comprehensive statistics of VG benchmarks: RefCOCO [Yu et al. (2016)], RefCOCO+ [Yu et al. (2016)], RefCOCOg [Nagaraja et al. (2016)], ReferIt [Kazemzadeh et al. (2014)], Flickr30k [Plummer et al. (2015)]. Test and testA are shown together.

| Datasets | Images | Annotated Instances | Avg. Text Length | Textual Queries | | | | |
| --- | --- | --- | --- | --- | --- | --- | --- | --- |
| | | | | total | train | val | test (A) | testB |
| RefCOCO | 19,994 | 50,000 | 3.49 | 142,210 | 120,624 | 10,834 | 5,657 | 5,095 |
| RefCOCO+ | 19,992 | 49,856 | 3.58 | 141,564 | 120,191 | 10,758 | 5,726 | 4,889 |
| RefCOCOg-u | 25,799 | 49,822 | 8.47 | 95,010 | 80,512 | 4,896 | 9,602 | - |
| RefCOCOg-g | 26,711 | 54,822 | 8.46 | 104,560 | 85,474 | 9,536 | - | - |
| ReferIt | 20,000 | 19,987 | 3.45 | 120,072 | 54,127 | 5,842 | 60,103 | - |
| Flickr30k | 31,783 | 427,000 | 1.59 | 456,107 | 427,193 | 14,433 | 14,481 | - |

Table 8: Comparison with pre-trained models on RefCOCO, RefCOCO+, and RefCOCOg datasets. Parameter counts include head and decoder only. Best results are shown in **bold**, and second-best are underlined.

| Methods | Visual Encoder | Params (M) | Pre-train Images | RefCOCO | | | RefCOCO+ | | | RefCOCOg | |
| --- | --- | --- | --- | --- | --- | --- | --- | --- | --- | --- | --- |
| | | | | val | testA | testB | val | testA | testB | val-u | test-u |
| UNITERL [Chen et al. (2020)] | RN101 | - | 4.6M | 81.41 | 87.04 | 74.17 | 75.90 | 81.45 | 66.70 | 74.86 | 75.77 |
| VILLAL [Gan et al. (2020)] | RN101 | - | 4.6M | 82.39 | 87.48 | 74.84 | 76.17 | 81.54 | 66.84 | 76.18 | 76.71 |
| MDETR [Kamath et al. (2021)] | RN101 | 17.36 | 200K | 86.75 | 89.58 | 81.41 | 79.52 | 84.09 | 70.62 | 81.64 | 80.89 |
| RefTR [Li & Sigal (2021)] | RN101 | 17.86 | 100K | 85.65 | 88.73 | 81.16 | 77.55 | 82.26 | 68.99 | 79.25 | 80.01 |
| SeqTR [Zhu et al. (2022)] | DN53 | 7.90 | 174K | 87.00 | 90.15 | 83.59 | 78.69 | 84.51 | 71.87 | 82.69 | 83.37 |
| UniTAB [Yang et al. (2022b)] | RN101 | - | 200K | 88.59 | 91.06 | 83.75 | 80.97 | 85.36 | 71.55 | 84.58 | 84.70 |
| DQ-DETR [Liu et al. (2023b)] | RN101 | - | 200K | 88.63 | 91.04 | 83.51 | 81.66 | 86.15 | 73.21 | 82.76 | 83.44 |
| GroundingDINO [Liu et al. (2024)] | Swin-T | - | 200K | 89.19 | 91.86 | 85.99 | 81.09 | 87.40 | 74.71 | 84.15 | 84.94 |
| PolyFormer [Liu et al. (2023a)] | Swin-B | - | 174K | 89.73 | 91.73 | 86.03 | 83.73 | 88.60 | 76.38 | 84.46 | 84.96 |
| PolyFormer [Liu et al. (2023a)] | Swin-L | - | 174K | 90.38 | 92.89 | 87.16 | 84.98 | 89.77 | 77.97 | 85.83 | 85.91 |
| OFA-L [Wang et al. (2022)] | RN152 | - | 20M | 90.05 | 92.93 | 85.26 | 85.80 | 89.87 | 79.22 | 85.89 | 86.55 |
| mPLUG-2 [Xu et al. (2023)] | ViT-L/14 | - | 14M | 92.40 | 94.51 | 88.42 | 86.02 | 90.17 | 78.17 | 85.88 | 86.42 |
| SimVG-DB [Dai et al. (2024)] | ViT-B/32 | 6.32 | 28K | 90.98 | 92.68 | 87.94 | 84.17 | 88.58 | 78.53 | 85.90 | 86.23 |
| SimVG-TB [Dai et al. (2024)] | ViT-L/32 | 1.58 | 28K | 92.99 | 94.86 | 90.12 | 87.43 | 91.02 | 82.10 | 87.95 | 88.96 |
| SimVG-DB [Dai et al. (2024)] | | 6.32 | 28K | 92.93 | 94.70 | 90.28 | 87.28 | 91.64 | 82.41 | 87.99 | 89.15 |
| ASVG-D | ViT-B/32 | 11.50 | 28K | 91.64 | 92.99 | 88.38 | 85.10 | 88.67 | 79.66 | 85.97 | 86.89 |
| ASVG-T | ViT-L/32 | 1.15 | 28K | **93.24** | **95.04** | **90.53** | **88.03** | 91.32 | 82.72 | 88.21 | 89.29 |
| ASVG-D | | 11.50 | 28K | 93.04 | 94.87 | 90.45 | 87.89 | **91.79** | **82.80** | **88.23** | **89.32** |

### A.3 Additional Implementation Details

This section provides additional details on the experimental setup. Apart from the encoder initialized from BEiT-3 pretrained weights, all other parameters are randomly initialized using PyTorch's default scheme. The Adam optimizer is adopted with a base learning rate of 5e-4, adjusted by a multi-step decay with linear warm-up. For the BEiT-large encoder, the batch size is set to 3 due to memory constraints. The encoder output dimension is expanded from 768 to 1024, with the threshold fixed at $\theta = 0.75$, while all other settings remain identical to those of the base encoder. In the pre-training experiments (Table 8), the results for the BEiT-base encoder are obtained from the decoder branch supervised by ground truth, whereas the BEiT-large encoder is still trained via online distillation with the SCD strategy. The number of training epochs is reduced to 50 for pretraining, further reduced to 40 for the large encoder due to the high computational cost, and uniformly set to 15 for fine-tuning. Unless otherwise specified, the distillation process adopts $\tau = 2$ and $\theta = 0.8$ as default hyperparameters. In GREC experiments, the number of object queries is fixed to 10, $\theta$ is set to 0.75, and the heatmap alignment loss coefficient $\lambda$ to 0.8 (Eq. 11). In other comparative experiments, $\theta$ is set to 0.8 and $\lambda$ to 1. Their influence is examined in the ablation study. Exponential Moving Average (EMA) is not applied during any stage of training. In addition, the pretrained model BEiT-3 is not trained on any of the six datasets used for evaluation. All results reported for both ASVG-T and ASVG-D are obtained through online distillation with SCD.

### A.4 Comparison with Pre-trained Models

Table 8 shows a comparison between ASVG and several pre-trained VG methods on the RefCOCO series datasets. With only 28K images used for pre-training, ASVG achieves performance comparable to or even exceeding methods trained on much larger datasets. ASVG-T (ViT-L) achieves 95.04% on RefCOCO testA and 90.53% on testB, and it surpasses most large-scale pre-trained methods such as GroundingDINO [Liu et al. (2024)] and PolyFormer [Liu et al. (2023a)] on the RefCOCO+/g datasets. Compared with SimVG [Dai et al. (2024)], ASVG consistently outperforms it across multiple datasets, further validating the effectiveness of introducing AP-QG and the progressive cross-scale decoder. In addition, ASVG maintains a lightweight architecture. ASVG-T contains only about 1.15M parameters, which is significantly smaller than some existing lightweight models while maintaining strong performance. Notably, when the cross-modal encoder is scaled from ViT-B to ViT-L, the lightweight token branch not only matches but in some cases exceeds its teacher (PCD). These results indicate that as the encoder size increases, the supervision provided by PCD becomes more reliable, thereby enhancing the generalization ability of the token branch.

### A.5 Additional Ablation Studies

***Ablations for Object Query Generation.*** In visual grounding, object queries are developed to represent the feature representation of the referred object. Early methods [Deng et al. (2021); Yang et al. (2022a); Deng et al. (2023)] generate queries from randomly initialized learnable vectors, which lack semantic relevance, making subsequent target reasoning difficult. Recent methods attempt to generate queries through text guidance, providing target-related prior context but overlooking the essential role of visual information in target semantic understanding. As shown in Figure 4 and Table 9, we take randomly initialized learnable vectors as the baseline. Compared with the Text-guided Query Generation (TQG) module proposed in [Dai et al. (2024)], our AP-QG exploits alignment priors from the encoder to construct target heatmaps and explicitly embed them into object queries, resulting in significant improvements in both convergence efficiency and final performance.

***Effect of Hyperparameters in Adaptive Loss Reweighting.*** We examine the effects of the threshold $\theta$ and temperature $\tau$ in the adaptive loss reweighting scheme (Eq. 10). Table 10 shows consistent trends for both the token branch and PCD. Setting $\theta = 0.8$ yields the best performance on test-u, improving by 1.00/0.94 percentage points over $\theta = 0.0$, while increasing $\theta$ to 0.9 causes a slight decline. These findings suggest enabling adaptive reweighting once the teacher has stabilized to avoid a late start that would shorten the effective window of reweighting. For $\tau$, smaller values produce overly peaked weights and hinder the collaboration of different losses, whereas larger values average them and weaken guidance from dominant terms. For example, $\tau = 5$ is slightly higher on val-u but underperforms $\tau = 2$ on test-u. The small gain on val-u likely reflects over-smoothing and local fitting. We therefore adopt $\theta = 0.8$ and $\tau = 2$ as the default configuration.

Table 9: Ablation Study on Object Query Generation. Each cell reports Token / Decoder accuracy (%).

| Methods | RefCOCO | | |
|---|---|---|---|
| | val | testA | testB |
| Baseline | 87.28 / 87.29 | 89.17 / 89.30 | 82.94 / 83.11 |
| TQG | 87.33 / 87.34 | 89.96 / 90.01 | 84.23 / 84.22 |
| AP-QG | 87.88 / 87.86 | 90.10 / 89.99 | 84.41 / 84.45 |

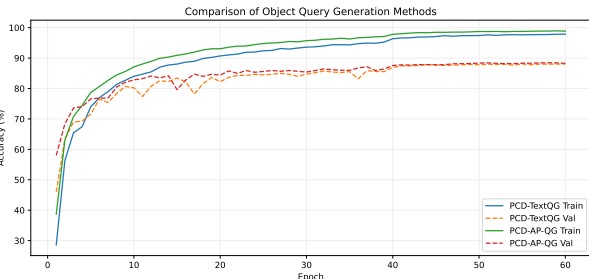

Figure 4: Comparison of accuracy over epochs for different object query generation methods.

Table 10: Effect of temperature $\tau$ and threshold $\theta$. Each cell reports Token / Decoder accuracy (%).

| Type | RefCOCOg | |
|---|---|---|
| | val-u | test-u |
| $\tau = 2$ ($\theta$ varies) | | |
| 0.0 | 79.62 / 79.78 | 80.48 / 80.65 |
| 0.4 | 79.77 / 80.09 | 80.18 / 80.26 |
| 0.8 | 80.06 / 80.48 | 81.48 / 81.59 |
| 0.9 | 80.52 / 80.69 | 80.72 / 80.97 |
| $\theta = 0.8$ ($\tau$ varies) | | |
| 0.5 | 80.46 / 80.50 | 81.03 / 81.13 |
| 1 | 80.22 / 80.12 | 80.94 / 81.02 |
| 2 | 80.06 / 80.48 | 81.48 / 81.59 |
| 5 | 81.11 / 81.19 | 81.20 / 81.29 |

***Analysis of the Heatmap Alignment Loss.*** As shown in Table 11, adding the heatmap alignment loss $\mathcal{L}_{\text{HA}}$ yields consistent gains for both the token branch and PCD on RefCOCO, showing that an explicit spatial alignment constraint enhances the model's perception of the referred object and thereby improves grounding performance. Table 12 further shows that the weight $\lambda$ for $\mathcal{L}_{\text{HA}}$ has a noticeable impact on performance: $\lambda = 1.0$ gives the best results, suggesting a good balance between KL supervision and the grounding loss. A properly chosen $\lambda$ improves both heatmap quality and localization accuracy, whereas an excessively large weight sacrifices grounding performance.

Table 11: Ablation of the heatmap alignment loss. Each cell reports Token / Decoder accuracy (%).

| Methods | RefCOCO | | |
|---|---|---|---|
| | val | testA | testB |
| w/o $\mathcal{L}_{\text{HA}}$ | 87.22 / 87.32 | 89.08 / 89.08 | 83.51 / 83.60 |
| w $\mathcal{L}_{\text{HA}}$ | 87.88 / 87.86 | 90.10 / 89.99 | 84.41 / 84.45 |

Table 12: Effect of the weight $\lambda$ for $\mathcal{L}_{\text{HA}}$. Each cell reports Token / Decoder accuracy (%).

| Type | RefCOCO | | |
|---|---|---|---|
| | val | testA | testB |
| 0.5 | 87.36 / 87.39 | 89.71 / 89.77 | 84.28 / 84.34 |
| 1.0 | 87.88 / 87.86 | 90.10 / 89.99 | 84.41 / 84.45 |
| 2.0 | 88.02 / 87.97 | 89.59 / 89.61 | 84.24 / 84.28 |
| 5.0 | 87.89 / 87.82 | 90.03 / 90.05 | 84.24 / 84.27 |

## A.6 TRAINING EFFICIENCY

Under the same hardware setup (4×RTX 3090), we compare each method's convergence cost (epochs) and wall-clock training time on RefCOCOg against the performance reported in the original papers. As shown in Table 13, ASVG, trained in a single stage, reaches 80.48%/81.59% (val-u/test-u) in 60 epochs/7.5h, requiring fewer epochs than VLTVG and Dynamic MDETR (both 90 epochs), indicating faster convergence. SimVG completes its first stage in 30 epochs / 4.5h, but its reported performance relies on a second-stage distillation (+20 epochs), so the overall training cost is not lower than ASVG. In sum, using original-paper metrics for fairness, ASVG achieves comparable or better accuracy-efficiency trade-offs with fewer epochs and a simpler single-stage pipeline.

Table 13: Training time on RefCOCOg (4×RTX 3090) vs. performance reported in the original papers. "Two-stage Training" denotes the presence of an additional training stage.

| Methods | Visual Encoder | Epoch | Training Time | Two-stage Training | RefCOCOg val-u | test-u |
|---|---|---|---|---|---|---|
| VLTVG | RN101 | 90 | - | × | 76.04 | 74.18 |
| SimVG$_{DB}$ | ViT-B/32 | 30 | 4.5h | ✓ | 80.37 | 80.51 |
| Dynamic MDETR | ViT-B/16 | 90 | - | × | 74.14 | 74.49 |
| ASVG-D | ViT-B/32 | 60 | 7.5h | × | 80.48 | 81.59 |

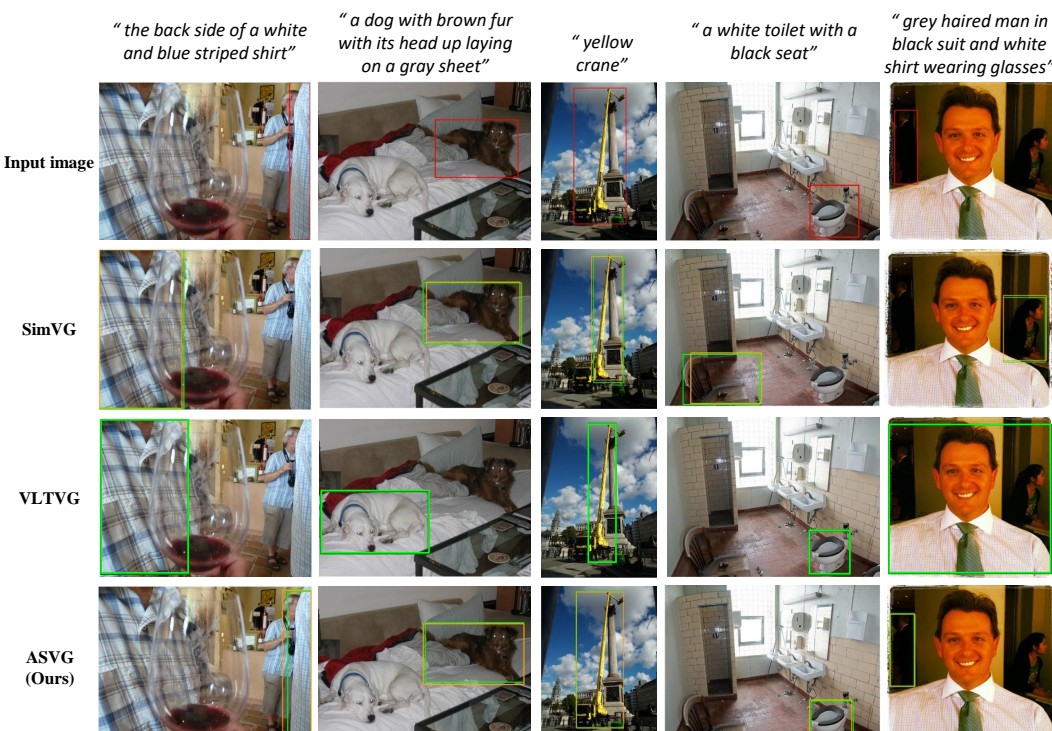

Figure 5: Examples predicted by ASVG on the validation set of the RefCOCOg dataset.

## A.7 SUPPLEMENTARY QUALITATIVE RESULTS

Table 5 presents a qualitative comparison between the proposed ASVG and representative VG methods (SimVG [Dai et al. (2024)] and VLTVG [Yang et al. (2022a)]). Red boxes denote ground-truth annotations, while green and orange boxes indicate predictions from the PCD and token branch, respectively. Overall, ASVG demonstrates more robust alignment with the described objects across diverse scenarios. In the first and fifth examples, other methods tend to extend predictions to visually similar foreground objects, whereas ASVG focuses only on the intended regions. In the second example, under a description with mild ambiguity, ASVG accurately localizes the brown dog with well-defined boundaries, while SimVG incorrectly predicts the white dog. In the third example, other methods often truncate elongated structures or include background regions, while our ASVG successfully captures the entire target, reflecting stronger cross-scale consistency. The token branch follows the same trend as PCD, achieving prediction quality that closely approaches the teacher.

Figure 6 shows qualitative results of ASVG on the gRefCOCO testA and testB splits. Red boxes is ground-truth annotations, and green and orange boxes indicate predictions from the PCD and token branch. The first row presents successful cases, showing that AP-QG exploits alignment priors to generate discriminative object queries and accurately localize multiple targets in complex scenes. For example, ASVG correctly identifies targets under explicit descriptions such as "blue umbrella" and "hands with camera." The second and third rows present failure cases. For multi-target or ambiguous descriptions, ASVG still struggles with boundary reasoning and fine-grained semantics.

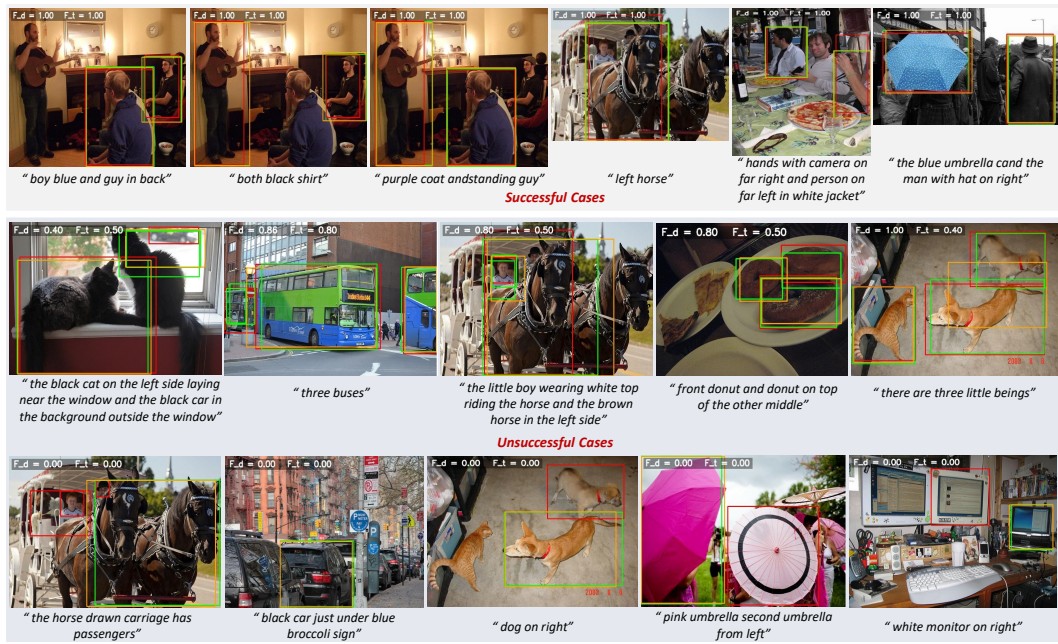

Figure 6: Examples predicted by ASVG on the testA and testB splits of the gRefCOCO dataset.

In the case of "the little boy wearing white top riding horse and the brown horse in the left side", multiple entities and attributes lead to partial misalignment, showing that multi-object interactions and complex attribute bindings remain challenging. For abstract expressions such as "three little beings," ASVG fails to distinguish instances, producing unstable predictions. These results indicate that ASVG achieves strong localization in most common, semantically clear scenarios, yet challenges remain in cases that require fine-grained reasoning over multiple objects. These findings confirm the effectiveness of incorporating image-text alignment priors through AP-QG and highlight future directions in multi-target grounding and fine-grained semantic modeling.

