# OpenReview forum: "Efficient Visual Grounding via Alignment Priors and Scale Adaptability"
_ICLR.cc/2026/Conference — ICLR 2026 Conference Withdrawn Submission_

### Official Review · Reviewer_LX9h · 2025-10-22

**Soundness:** 3
**Presentation:** 3
**Contribution:** 2
**Rating:** 6
**Confidence:** 4

**Summary:**

This paper proposes ASVG, a novel framework for visual grounding (VG) and generalized referring expression comprehension (GREC).
he key ideas are:
An Alignment Prior-guided Query Generator (AP-QG) that leverages cross-modal alignment priors from pretrained encoders  to produce target-aware queries.  A Progressive Cross-scale Decoder (PCD) that performs multi-resolution reasoning using only single-scale features, avoiding redundant feature fusion. A Soft Cross-head Distillation (SCD) strategy and a lightweight token branch to preserve performance while improving inference efficiency.

**Strengths:**

The paper identifies two under-explored bottlenecks in VG and proposes targeted solutions. The idea of explicitly using alignment priors from pretrained encoders to construct queries is both conceptually clean and practically effective.

The proposed Progressive Cross-scale Decoder (PCD) avoids FPN-style heavy multi-scale fusion while maintaining multi-resolution reasoning ability. The design is lightweight, and experiments confirm substantial efficiency gains.

**Weaknesses:**

The alignment-prior concept largely reuses the cross-modal similarity from pretrained encoders (BEiT/CLIP). Similar concepts (text-conditioned visual attention, saliency-weighted queries) already appear in HiVG and SimVG. The novelty seems implementation-level rather than conceptual.

ASVG heavily relies on strong pretrained cross-modal encoders like BEiT-3 or CLIP. It’s unclear how the method performs without such pretrained priors or under domain shift conditions where the pretrained model’s alignment space is weak.

While comparisons include recent works (SimVG, HiVG, CLIP-VG), the pretraining data size and visual backbone differences (e.g., ViT-L/32 vs ViT-B/16) could confound the performance gap. More controlled ablations (same backbone across baselines) would strengthen the claim.

**Questions:**

Could the authors clarify how the AP-QG differs mathematically from the text-guided query initialization in SimVG? Specifically, does it only add cross-modal heatmap weighting, or also modifies the feature projection space?

Since the model heavily relies on pretrained BEiT-3 encoders, how does ASVG perform when using smaller encoders (e.g., ViT-B/16 or CLIP-B)?

In Eq. (9–10), the adaptive loss reweighting seems heuristic. Could the authors explain how the weights are learned or updated during training?

Is the PCD jointly optimized with the encoder, or are the scale levels frozen after pretraining?

---

### Official Review · Reviewer_gKGQ · 2025-10-31

**Soundness:** 2
**Presentation:** 3
**Contribution:** 2
**Rating:** 6
**Confidence:** 4

**Summary:**

This paper introduces ASVG, an efficient framework for Visual Grounding that aims to solve two problems: semantic alignment bias and scale-induced perception mismatch. The authors propose an Alignment Prior-guided Query Generator (AP-QG) that leverages the cross-modal encoder's intrinsic alignment to create target-aware object queries. To handle objects of various sizes efficiently, they design a Progressive Cross-scale Decoder (PCD) that constructs a feature pyramid from a single-scale input. Finally, they introduce a lightweight "token branch" trained with a Soft Cross-head Distillation (SCD) strategy to provide a fast inference path. Experiments across six VG and GREC datasets show that ASVG achieves state-of-the-art or highly competitive performance with significant efficiency gains.

**Strengths:**

1. The core innovation, the Alignment Prior-guided Query Generator (AP-QG), is intuitive. By embedding a text-conditioned visual heatmap into the object queries, it directly addresses the semantic alignment bias and provides the decoder with a much stronger starting point for reasoning.

2.  The Progressive Cross-scale Decoder (PCD) avoids the computational overhead of a full feature pyramid network, and the addition of the distilled token branch provides a valuable, lightweight deployment option.

3. The method is supported by extensive experiments. The authors validate their method on a wide range of datasets and provide thorough ablation studies that clearly demonstrate the effectiveness of each proposed component.

**Weaknesses:**

1.  While ASVG demonstrates superior performance and efficiency on benchmarks, the paper should explicitly argue for its relevance by highlighting advantages in areas LMMs might lack, such as deployment cost, inference speed.

2. The proposed components, such as the Progressive Cross-scale Decoder (PCD) and Soft Cross-dead Distillation (SCD, is somewhat incremental, which means the design is not impressive enough to inspire further researches.

Minor:
1. Regarding the failure cases, could you provide a more in-depth analysis of the potential sources of error and give insights about how to solve it?

**Questions:**

Please see the Weaknesses.

---

### Official Review · Reviewer_cUTm · 2025-11-01

**Soundness:** 2
**Presentation:** 3
**Contribution:** 2
**Rating:** 4
**Confidence:** 4

**Summary:**

This work proposes ASVG, an efficient visual grounding framework that combines alignment priors with cross-scale reasoning to strengthen the semantic discriminability of queries and handle targets at varying scales. Its core components are AP-QG, which leverages encoder alignment priors to produce target-aware queries, and PCD, which builds a multi-resolution pyramid from single-scale features to avoid redundant fusion; in addition, a lightweight token branch with SCD enforces feature consistency and adaptively reweights losses to reduce inference cost. Extensive experiments on six mainstream VG and GREC datasets validate the effectiveness and efficiency of ASVG.

**Strengths:**

1.The paper couples alignment-prior–guided query initialization (AP-QG) with a progressive cross-scale decoder (PCD) that builds a pyramid from single-scale features, avoiding heavy FPN-style fusion. This is a clean way to mitigate semantic-alignment bias and improve scale adaptability with lower computational overhead.

2.The method achieves consistent performance improvements across six benchmark datasets while simultaneously reducing inference cost.

**Weaknesses:**

1. In the AP-QG module, Eq.3 introduces a semantic discontinuity and reads as heuristic:

(1) The transformation from the spatial response map M_heatmap to the query vector Q_init introduces a semantic discontinuity . M_heatmap contains scalar spatial activations rather than semantic features; projecting these scalars directly into the query space risks breaking the semantic link between visual evidence and textual meaning. Without explicitly aggregating underlying visual tokens (e.g., from F_v) before forming Q_init, it is unclear how the resulting query can encode object-level semantics or spatial context in a principled way.

(2) A linear projection of a one-dimensional heatmap added to a pooled text vector does not explain why the resulting embedding should be discriminative or robust. More principled alternatives—such as heatmap-weighted aggregation of visual tokens followed by concatenation or gated fusion, or a lightweight cross-attention—would better preserve cross-modal correspondence while remaining efficient.

2. Some unclear description about the progressive cross-scale decoder (PCD):

(1) The manuscript claims that the ConvT-based upsampling “approximates Shannon reconstruction” and that the 3×3 convolution “approximates a discrete Laplacian,” but this is asserted without evidence. The authors are required to provide a detailed explanation and supporting analysis to justify these claims.

(2) To substantiate the benefit of PCD, please add an ablation that replaces PCD with a standard FPN and report the accuracy and the inference time.

(3) In Table 4, the first three rows exhibit abnormally low performance on small objects, especially the third row (“w/o 1/64 → 1/32”), where removing only the deepest/coarsest scale should primarily affect large-object context rather than devastate small-object accuracy (16.67/16.67). This is counter-intuitive. The authors are required to provide a detailed explanation about this.

3. Some grammatical and expression issues:
(1) Typo in the introduction: “Cross-dead” should be “Cross-head”, and not just once.

(2) Figure 2 legend/semantics: The meanings of the different colored arrows are unclear and must be explicitly explained in the caption and/or figure legend.

(3) Model choice rationale: Why BEiT-3 is more suitable than CLIP for this task needs a brief justification.

**Questions:**

Please refer to the weakness part.

---

### Official Review · Reviewer_LXYv · 2025-11-04

**Soundness:** 2
**Presentation:** 2
**Contribution:** 2
**Rating:** 2
**Confidence:** 5

**Summary:**

See Questions.

**Strengths:**

See Questions.

**Weaknesses:**

See Questions.

**Questions:**

Review Comments:

After reading the manuscript, I have the following comments. I hope the authors could address them carefully.

- Q1. In Figure 1(a), the direction of the arrow is incorrect. It should indicate that the language guides the visual backbone, not the other way around.

- Q2. The literature review in this paper is comprehensive, which suggests that the authors have an understanding of the field. However, OneRef (NeurIPS 2024) [1] is also a strong model based on BEiT-3, and its performance surpasses that of SimVG. This work should include a discussion and comparison with OneRef.

- Q3. The acronym ASVG is not introduced or explained in the introduction, which may confuse readers unfamiliar with the term.

- Q4. The pipeline presented in Figure 2 is overly complex and difficult to interpret. For example, the meaning of the green, blue, and red lines is not explained. It is unclear why the AP-QG module receives inputs from both the blue and green lines, and why there is a connection from the red line to the black line. Additionally, the top portion of the figure shows another AP-QG-to-PCD/SCD branch, which could be misinterpreted as an extra computational path. This figure needs to be simplified and clarified.

- Q5. The Related Work section spends too much space discussing outdated one-stage and two-stage methods. These approaches are no longer the focus of current state-of-the-art research and could be condensed to make room for the discussion of more recent work.

- Q6. In Section 3.2, the query generation mechanism described is quite common and has already been widely explored in numerous previous works. For example, VLT[2]. The paper should clarify what is novel about the proposed approach, if anything.

- Q7. Similarly, the use of multi-scale visual information in Section 3.3 is not a novel idea. This concept has been widely adopted in many detection-based models. The paper should highlight what makes their use of multi-scale features different or innovative.

- Q8. The one-stage and two-stage methods listed in Table 1 are outdated. For a top-tier conference paper, it is no longer necessary to include such baselines. Instead, the authors should focus on more recent and competitive approaches. In particular, Table 1 should include comparisons with other BEiT-3-based methods, such as OneRef, to better contextualize the performance of the proposed model.

---

[1] OneRef: Unified One-tower Expression Grounding and Segmentation with Mask Referring Modeling. NeurIPS 2024.

[2]  Vision-Language Transformer and Query Generation for Referring Segmentation. ICCV 2021.

---

### Note · Authors · 2025-11-15

I have read and agree with the venue's withdrawal policy on behalf of myself and my co-authors.